# Multisystem Amyloidosis in a Coal Miner with Silicosis: Is Exposure to Silica Dust a Cause of Amyloid Deposition?

**DOI:** 10.3390/ijerph19042297

**Published:** 2022-02-17

**Authors:** Tomasz Gołębiowski, Jakub Kuźniar, Tomasz Porażko, Renata Wojtala, Andrzej Konieczny, Magdalena Krajewska, Marian Klinger

**Affiliations:** 1Department of Nephrology and Transplantation Medicine, Wroclaw Medical University, 50-556 Wroclaw, Poland; klinef@am.centrum.pl (J.K.); andrzej.konieczny@umw.edu.pl (A.K.); magdalena.krajewska@umw.edu.pl (M.K.); 2Department of Internal Medicine and Nephrology, Institute of Medical Sciences, University of Opole, 45-052 Opole, Poland; tporazko@uni.opole.pl (T.P.); klinger@wp.pl (M.K.); 3Departament of Patomorphology, Wroclaw Medical University, 50-556 Wroclaw, Poland; drwojtala@gmail.com

**Keywords:** silicosis, primary amyloidosis, heart insufficiency, kidney insufficiency, proteinuria

## Abstract

The over-secretion of monoclonal immunoglobulin light chains by clonal B cells followed by the aggregation and extracellular deposition of fibrillar deposits are responsible forthe clinical course AL amyloidosis. It is well documented that silica significantly increases the number of immunoglobulin-secreting cells. In the present paper, we report on a coal miner with silicosis and fast progressing primary amyloidosis with predominantly heart, kidney, and lung manifestations. Severeheart failure due to myocardial hypertrophy resulted in the patient’s death. We conclude that long-term environmental silica exposure and silica deposition may contribute to the development of monoclonal gammopathy and amyloidosis due to chronic stimulus and the dysregulation of the immune system.

## 1. Introduction

The correlation between dust exposure (silica) and silicosis has been well documented in medical literature. Acute silicosis involves a severe immunological reaction in lung tissue, which results in the destruction of its structure. However, chronic silicosis involves the formation ofgranuloma and lung tissue fibrosis. Chronic silicosis develops over an extended period of time and results in restrictive respiratory insufficiency. Symptoms may appear a long time after silica exposure. Studies show an association between chronic silicosis and autoimmune disease, such as systemic sclerosis, lupus erythematosus, rheumatoid arthritis, dermatomyositis, and vasculitis [1,2,3,4,5,6]. Epidemiological studies among coal miners with silicosis have demonstrated a high frequency of antinuclear antibodies and rheumatoid factor [7]. Exposure to silica may also be the prime factor of glomerulonephritis. Kolev et al. showed focal segmental glomerulonephritis in 23 cases, in a group of 45 patients with silicosis [8]. Saldanha et al. reported a case of 44 males with proteinuria and hypertension linked to silica dust exposure. A renal biopsy showed focal segmental glomerulonephritis, degenerative changes in renal tubules and a high level of silicon in the kidney [9]. Additionally, reports have shown a correlation between silica exposure and Goodpasture syndrome [10], fast progressing glomerulonephritis with crescents [11], IgA nephropathy with crescents [12]. One Italian study on ceramic industry employees demonstrated an increased incidence of end-stage renal disease linked to dust exposure [13].

In the present paper, we report on a coal miner suffering from silicosis who developed fast progressive multisystem amyloidosis. We provide a detailed clinical history and complete histological microscope picture of the condition. To our knowledge, this is probably the first report that examines the link between silicosis and amyloidosis. In the Section 4, based on contemporary animal studies, we show immunopathological disorders related to silica dust exposition and its association with plasma cell dyscrasia.

## 2. Case Report

A 52-year-old male with a 9-year record of diagnosed silicosis (with 24 years of occupational silica dust exposurein rock blasting) was admitted to the Nephrology Department because of proteinuria. Urine protein loss had been observed for two years, initially 50–150 mg per day and 3.5 g per day at the time of admission. In addition, the progressive impairment of kidney function was observed, with a creatinine level of 1.9 mg/dL (eGFR (MDRD) 40 mL/min/1.73 m^2^) 5 months before and 3.3 mg/dL (eGFR (MDRD) 21 mL/min/1.73 m^2^) at the admission time.

He had been treated for 12 years for hypertension. During the last few years, he developed paroxysmal atrial fibrillation. A year before his admission, he suffered a transient ischemic stroke. For 6 months, bilateral hydrothorax had been observed, with the right-side predominance, requiring repeated puncture.

The patient was admitted with stable circulatory and respiratory functions and a good mental state. A clinical examination revealed bilateral hydrothorax and ankle swelling. Chest computer tomography (CT) confirmed the presence of mediastinal lymphadenopathy, multiple small pulmonary nodules accompanied by calcifications, hydrothorax in both pleural cavities (Figure 1A–C).

The bronchoscopy revealed signs of chronic inflammation and restrictive ventilatory disorder in spirometry (volume capacity (VC) 34.0%, forced volume capacity (FVC) 32.1%, forced expiratory volume in 1 s (FEV1) 33.8%, FEV1/VC 97.8%). The transthoracic echocardiogram showed a normal left ventricle size with a good contractile function and mild mitral insufficiency. An abdominal ultrasonography revealed no abnormalities, except for a hyperechogenic kidney structure (right kidney 11.8 cm × 5.6 cm and left 11.9 cm × 5.6 cm).

Laboratory testing showed hemoglobin—11.2 g%; hematocrit—33%; platelets—265,000/mm^3^; leukocytes—6000/mm^3^ with normal blood smear; serum creatinine—3.3 mg% (GFR MDR 21 mL/min/1.73 m^2^); serum urea—13.6 mmol/L; uric acid—8.5 mg/dL; total protein—61.9 g/L; albumin—30 g/L; cholesterol—3.6 mmol/L; HDL—1.04 mmol/L; triglyceride—1.49 mmol/L; C-reactive protein (CRP)—12.2 mg/dL; and erythrocyte sedimentation rate (ESR) 67/1 h. A urine examination revealed proteinuria from 150 to 2014 mg/dL (2.1 to 8.19 g/per day), erythrocyturia 6–8/hpf, and single hyaline-granular casts. Connective tissue disorder markers were negative (ANA, ANCA, cardiolipine, nRNP, Sm, SS-A, SS-B, Sci-70, Jo-1 antibodies). Complement hemolytic activity was normal. Protein electrophoresis showed immunoglobulin IgG concentration—−11.3 g/L (Norm: 8–17 g/L); Kappa light chain—−1.03 g/L (Norm: 1.7–3.7 g/L); Lambda light chain—−3.52 g/L (Norm: 0.9–2.1 g/L); and Kappa to Lambda ratio—0.29 (Norm 1.35–2.65). The serum immunofixation showed the presence of a monoclonal protein in the IgG class, the lambda-type light chain in the gamma globulin fraction. The microbiological examination of pleural exudative fluid was negative. Pulmonary tuberculosis was excluded by Ziehl–Neelsen sputum smear microscopy and the culture of sputum, pleural fluid and bronchial washings in a Mycobacterium Growth Indicator Tube (MGIT) (BD BACTEC™ MGIT™ 960 System Becton, Dickinson & Company (BD), Franklin Lakes, NJ, USA). Conventional Lowenstein–Jensen cultures of these fluids were also negative.

An abdominal fat biopsy confirmed the diagnosis of primary amyloidosis AL with lambda light chains.

Hemodialysis therapy was started because of the patient’s renal function deterioration and fluid overload. Despite daily hemodialysis, the patient’s clinical condition deteriorated as he developed severe cardiac insufficiency with pulmonary edema in addition to unresolved hydrothorax, although he remained normovolemic. A brain CT was carried out to evaluate the neurological manifestation of the patient as he lapsed into an altered level of consciousness; however, no evidence of stroke was demonstrated. The death of the patient was preceded with bradyarrhythmia followed by electro-mechanical dissociation. The patient died after cardiac arrest despite reanimation (3 months after the first admission to our clinic).

## 3. Autopsy

An autopsy showed no single immediate cause of death. There were no signs of myocardial infarct or stroke. The examination revealed heart and liver hypertrophy, and bilateral hydrothorax. A high amount of amyloid deposits and birefringent particles were found in the small myocardial arteries (Figure 2A,B) and in the kidney, in glomeruli and focally in kidney interstitium (Figure 3A,B). The spatial localization of the amyloid deposition was consistent with the presence of birefringent particles.

The EnVision method was used in the immunohistochemical examination. The Examination showsthe expression of light chain lambda immunoglobulin and P protein (Figure 4), but no amyloid protein A, transthyretin, ß2-microglobulin, or kappa light chain. A Lung histology showssilica nodules with intense inflammation (Figure 5A,B).

## 4. Discussion

The development of the condition was typical neither for advanced silicosis nor for amyloidosis AL. It was characterized by an unusual dynamic of multiple organ disorders. Pathogenesis involved two factors—silica and amyloid deposition, as was shown by the typical symptoms. Recurrent hydrothorax that required frequent drainage can, in rare cases, be attributed to silicosis [14] as well as to amyloidosis AL [15].

Heart insufficiency and neurological disorders can be symptoms of both silicosis and amyloidosis. Only in exceptional cases can the rapid development of symptoms and their consequent amplified impact on our patient be attributed to a coincidence of two disorders.

The disease developed in two stages. Stage one involved silicosis, hypertension, and mild proteinuria and was spread over many years. Stage two, which lasted for about 2 years, was linked to amyloidosis and increased proteinuria (over 3.5 g).

The rapid development of symptoms took place during kidney function impairment. It progressed despite dialysis treatment, leading to the patient’s death. Clinical course and autopsy show that amyloidosis caused the last stage of disease, but coexisting silicosis may have played a double role: firstly, as an initiator of the process (stimulating lymphocyte B clones toproduce monoclonal immunoglobulin and light chain production) and secondly as an accelerator of pathological fibril production and their tissue deposition.

Silica particles are known to be an adjuvant factor; the substance activates the immunologic system [16]. Macrophages play a central role in this process. They absorb silica particles, which, in turn, cause spontaneous destruction and release of free proteolytic enzymes. This is followed by cytokine release, the stimulation of the humoral and cell immunologic system, and the proliferation of lymphocytes T and B. This is why silica-exposed persons developgeneralized immunologic system dysfunction, as shown by the high frequency of glomerulonephritis and autoimmunologic disorders (rheumatoid arthritis, scleroderma, and lupus erythematosus) [2,3,4,5,6].

The diagnosis of AL amyloidosis in our patient was based on the International Myeloma Working Group criteria [17] and included the presence of an amyloid-related systemic syndrome (e.g., renal and heart), positive amyloid staining by Congo red in fat aspirate and positive serum immunofixation. An autopsy confirmed amyloidogenic immunoglobulin light chain deposits of the kidney and heart.

The exact mechanism of how advanced silicosis causes monoclonal immunoglobulin production is not entirely known; however, experimental data has shown that silica may have played an important role in lymphocyte B clone generation and immunoglobulin production [18]. The chronic inflammatory state observed in patients with silicosis probably plays a key role in stimulating B cells as well [16]. Additionally, the specific tissue deposition pattern typical for patients with end-stage renal disease may play a role in transforming amyloid protein into an insoluble fibril structure [19].

Amyloidosis light chains can be created under laboratory conditions using proteolytic enzymes (pepsin). Souillact et al., using protein LEN (protein Bence Jones [BJ]κIV) in an amyloidogenesis model, proved that a high concentration of urea accelerates the process of fibril formation [20]. Additionally, Rostagno et al. showed in an in vitro experiment that the BJ protein adopted a fibrillar conformation only at acidic pHs, remaining aggregated but not fibrillar at physiological pH [21].

Overall, the evidence exists to supportthe claim thatmetabolic acidosis and a high concentration of urea had a fundamental role in accelerating amyloid formation in our patient, despite dialysis treatment [20].

Silica alone could have taken part in the acceleration of amyloidosis, as confirmed in the experimental research of Zhu et al. The authors demonstrated that fibril creation is more extensive on the surface. The study showed that mica, one of the granite components containing silica elements, provided a surface conducive to amyloid creation [22].

It should be mentioned that, apart from generalized amyloidosis, there are also data showing that asbestos dust can be associated with amyloidosis limited to lung tissue in the form of solitary pulmonary nodule. Hiroshima et al. described a well-documented case of a patient in whom amyloid deposits were located at the site of asbestos fiber deposition [23]. In addition, a case of a dental technician with pulmonary amyloidosis as a result of occupational exposure to silica was described [24]. These are further arguments confirming the amyloidogenic effect of dusts on the development of immune system disorders. Conversely, chronic inflammatory autoimmune disease, such as Sjögren’s syndrome [25] and rheumatoid arthritis [26], may be the cause of a localized or generalized pulmonary amyloidosis [27]. In the latter study, Hara et al. showed that 3 out of 8 patients with systemic Sjögren’s syndrome developed generalized pulmonary amyloidosis in combination with mucosa-associated lymphoid tissue (MALT) lymphoma. However, there are also cases of nodular pulmonary amyloidosis, which are not associated with any additional disorders and may only constitute a diagnostic problem in the context of differentiation from a lung cancer [28].

In the last stage of the disease, our patient developed symptoms of multiple organ dysfunctions. The autopsy presented no unequivocal cause of death. Our case study lends evidence that the patient’s death was caused by a cardiac arrest due to arrhythmia. The autopsy showed concentric myocardium hypertrophy and the significant narrowing of the small arteries due to amyloid deposition. This observation is consistent with other autopsy studies among patients with primary amyloidosis. Heart hypertrophy is caused by amyloid deposition in the myocardium, which leads to diastolic heart dysfunction and vasculopathy, which in turn leads to myocardial ischemia without atherosclerotic changes in coronary arteries [29,30,31]. The autopsy confirmed that this process led to the death of our patients. One last additional factor to be mentioned is the cardiotoxicity of light chains [32].

Naturally, we cannot rule out the independent development of AL amyloidosis in our patient, although the clinical data and the quoted laboratory data confirm this. We also cannot show with certainty the path of disease development in this patient and, therefore, our report rather opens the door to discussions and further research.

## 5. Conclusions

Our study is probably the first case of primary amyloidosis triggered by silicon exposition. Long-term environmental silica exposure and silica deposition may contribute to the development of monoclonal gammopathy and amyloidosis due to chronic stimulus and the dysregulation of the immune system. Hopefully, it opens a debate on an important research problem that has grave implications for treatment.

## Figures and Tables

**Figure 1 ijerph-19-02297-f001:**
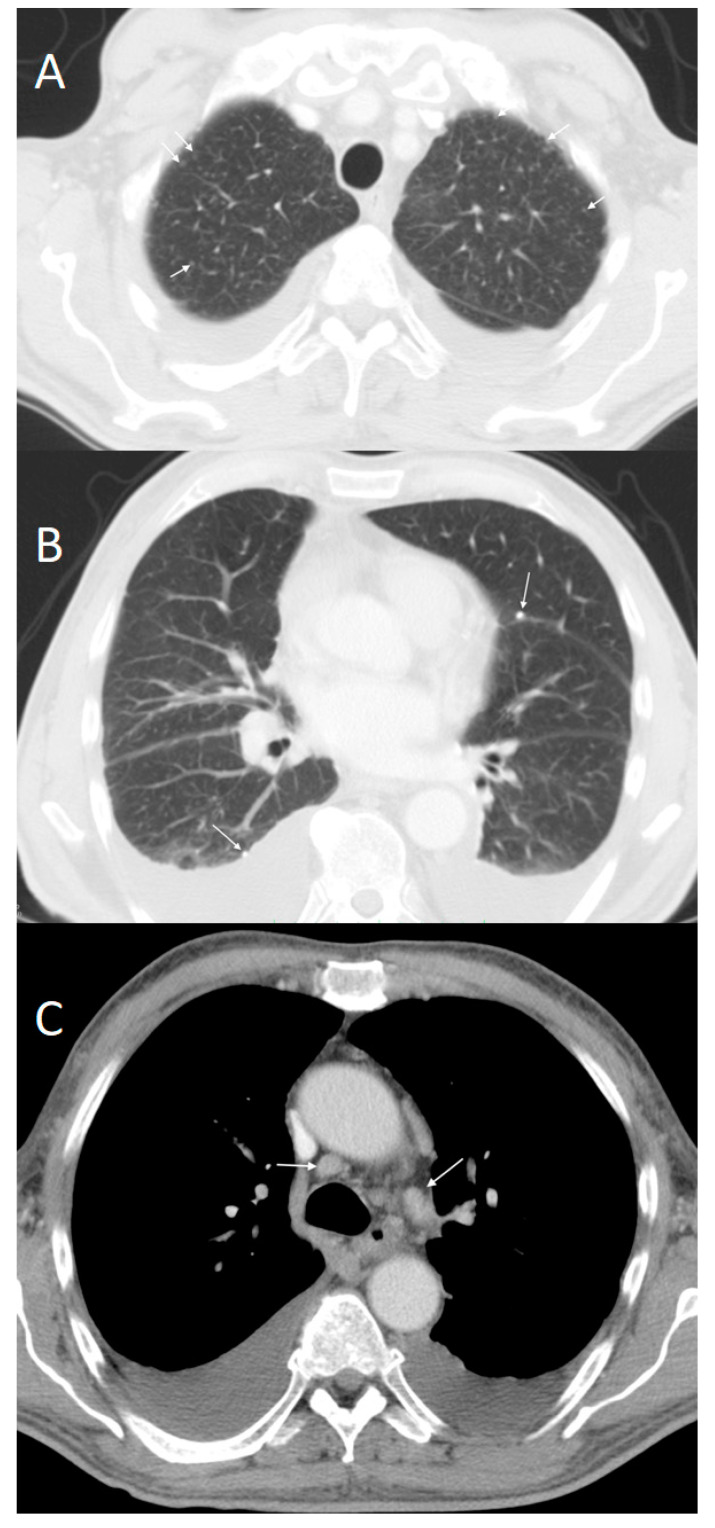
CT axial scans in pulmonary and soft tissue window revealed: (**A**) multiple small pulmonary nodules distributed predominantly in the upper lobes (white arrows); (**B**) nodules accompanied by calcifications (white arrows); and (**C**) mediastinal lymphadenopathy (white arrows) and bilateral hydrothorax.

**Figure 2 ijerph-19-02297-f002:**
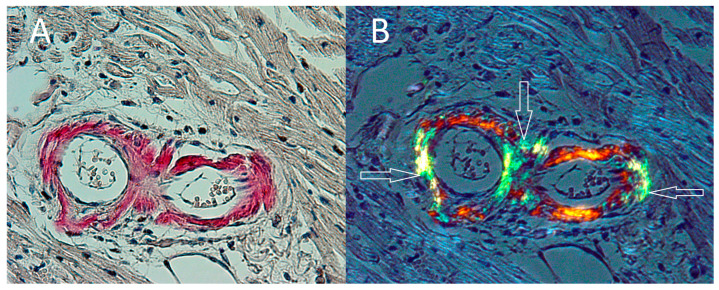
(**A**) Amyloid deposits in the small myocardial arteries. Magnification × 200. Congo red stain. (**B**) The same deposits in the arterial wall under polarized light (white arrows). Magnification × 200. Congo red stain.

**Figure 3 ijerph-19-02297-f003:**
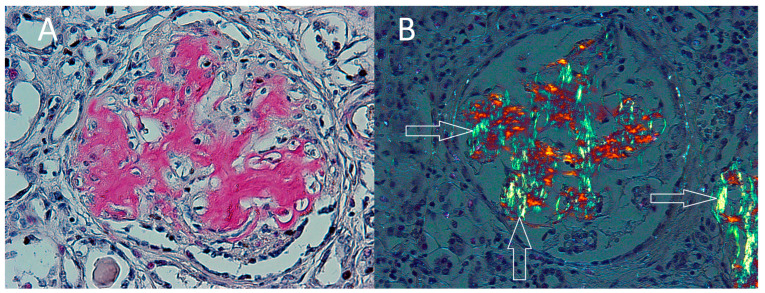
(**A**) Congo red positive deposits in glomerulus of the kidney. Magnification × 200. (**B**) Birefringent particles in the glomerulus and focally in the interstitium presenting a green apple color after staining with Congo red and seen under polarized light (white arrows). Magnification × 200.

**Figure 4 ijerph-19-02297-f004:**
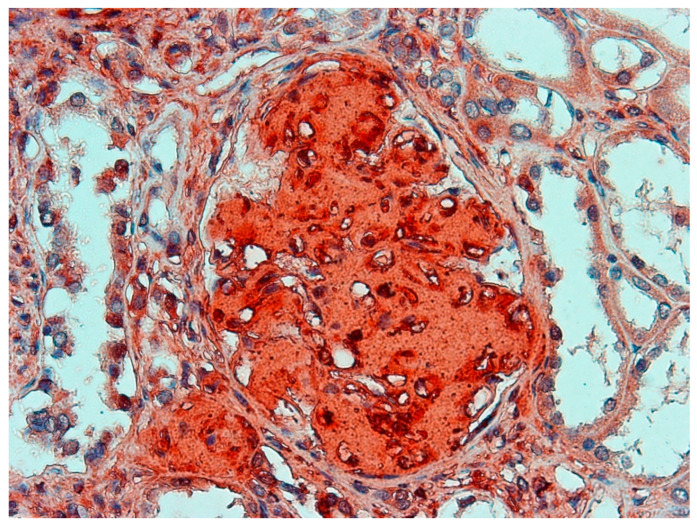
Expression of P protein in glomeruli of the kidney. Magnification × 200. EnVision stain.

**Figure 5 ijerph-19-02297-f005:**
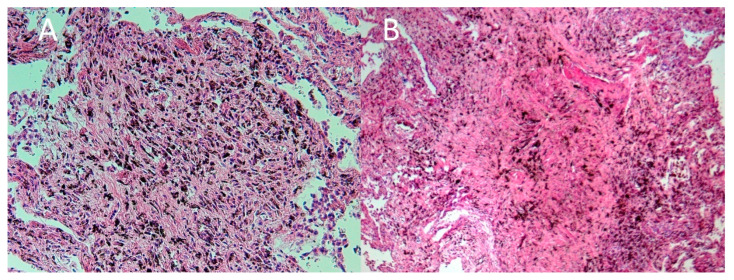
(**A**) Granuloma in the lung tissue with intense inflammation. Magnification × 100. Hematoxylin and eosin stain (H&E), (**B**) silica nodule with an amorphous substance and an inflammatory reaction surrounding it. Magnification × 100, H&E.

## Data Availability

The data presented in this study are available in this article.

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
