# Peer review of "Multisystem Amyloidosis in a Coal Miner with Silicosis: Is Exposure to Silica Dust a Cause of Amyloid Deposition?"

_ijerph, 2022, doi:10.3390/ijerph19042297_

Round 1

Reviewer 1 Report

comments to the author can be found in the manuscript edit section

Author Response

We appreciate important suggestions for text proofreading. The manuscript was additionally reviewed by the native speaker.

Reviewer 2 Report

This is a report of a single case of amyloidosis in a coal miner with silicosis.

This association has not been described previously but a number of case reports have been published (by Japanese authors) of a possible association with asbestos exposure. This literature should be mentioned and discussed.

A major problem with this report is the poor documentation of exposure to silica and silicosis: no chest imaging (CT), no mention of typical silicotic nodules in lung tissue, no evidence of birefringent particles (polarized light) in histology (lung tissue, myocardium, kidney?). The demonstration of a spatial relation between particles and amyloid deposition would substantially strengthen the argument. 

Moreover, the paper is very poorly written with poor grammar and numerous spelling errors.

Author Response

We thank you for this remarks.

At the reviewer's request, we provided additional literature and added the following text in the Discussion Section:

“It should be mentioned that apart from generalized amyloidosis, there are also data showing that asbestos dust can be associated with amyloidosis limited to lung tissue in the form of solitary pulmonary nodule. Hiroshima et al. describes a well-documented case of a patient in whom amyloid deposits were located at the site of asbezite fiber deposition [23]. In addition, a case of a dental technician with pulmonary amyloidosis as a result of occupational exposure to silica has been described [24]. These are further arguments enhancing the amyloidogenic effect of dusts on the development of immune system disorders. Conversely, chronic inflammatory autoimmune disease, such as Sjögren's syndrome [25] and rheumatoid arthritis [26], may be the cause of a localized or genera-lized pulmonary amyloidosis [27]. In the latter study, Hara et al. showed that 3 out of 8 patients with systemic Sjögren's syndrome developed generalized pulmonary amy-loidosis in combination with mucosa-associated lymphoid tissue (MALT) lymphoma. However, there are also cases of nodular pulmonary amyloidosis which are not asso-ciated with any additional disorders and may only constitute a diagnostic problem in the context of differentiation from a lung cancer [28].”

We agree with the reviewer that the original version of the manuscript contained little clinical information. It was related to the manuscript form, i.e. a Case Report in which only the most important clinical information was presented. The present text was supplemented with lung CT, kidney and heart histology in polarized light with spatial the relation between relationship between particles and amyloid deposition. Unfortunately, no histopathological examination of the Congo red stained silica nodule was found. We only have material stained with hematoxylin and eosin (H&E). In the manuscript we present two separate nodules: one with an intense inflammatory process, the other with an amorphous substance and an inflammatory reaction surrounding it.

Here, too, we must express our regret that the staining type in Figure 4 of the first manuscript was incorrectly signed. We are very sorry for the lack of scrutiny.

Nevertheless, we believe that the course of the disease was primarily associated with silicosis and chronic pulmonary inflammation - which is supported by post-mortem histopathological examination of the lungs and CT imaging. This dysregulation of the immune system led to the stimulation of B lymphocytes and, consequently, to generalized and multiorgan amyloidosis which was the cause of death.

The manuscript was additionally reviewed by the native speaker.

Reviewer 3 Report

This is very interesting case report.  Please have a first-language english speaker who understands the work to check the manuscript, particularly the case report section.  The language in the case report section currently is sometimes not clear, for example lines 54-59.  This is only an example, the whole case report section should have a careful edit to clarify the language.  Please provide information about informed consent and medical ethics (IRB) approval.

Author Response

Thank you for this opinion. The manuscript was reviewed by a native speaker. We have also modified the Informed Consent Statement according to the IJERPH rules.  

Round 2

Reviewer 2 Report

The authors have attempted to address my previous comments, but unfortunately they have only succeeded partially.

My main concern concerns the documentation of silicosis and dust particles.

  • the images on chest CT are not very convincing for silicosis
  • the histopathology of the lung (figure 5) is not typical for silicosis (necrotic granuloma?)
  • the demonstration of birefringent particles under polarized light (figures 2 and 3) needs more evidence

Moreover, the paper still contains a high number of inaccurate or wrong wording, grammatical and spelling errors, and typos; careless presentation does not encourage reviewers and readers to take the findings seriously.

Author Response

Response to the reviewer

We would like to thank the reviewer for his insightful contribution to our manuscript. We agree that the clinical data provided in the last version of our report do not sufficiently confirm the course of the patients disease. He has been evaluated and taken tissue samples several years ago. Therefore we are not able to perform additional histopathological tests of the tissue sample. 

However, we want to emphasize the following arguments supporting our view of the course of the disease:

First, the patient, was exposed to silica dust for over 20 years, working as mining shotfirer. That time, self-protection measures in Poland were not sufficient, if they were available at all. The patient was diagnosed with silicosis already in the 90's by occupational medicine service and has been granted with a disabled pension.

Secondly, the lungs CT scan revealed typical findings in the upper and middle parts described as the silicica nodules. Additionally, the findings were confirmed as silicosis by the pathologist on the autopsy. Moreover the inflammatory changes have been found in the tissue silica depositions with and also reflected by the mediastinal lymphadenopathy. Thorough clinical workout has excluded other causes of lung lesions as vasculitis (negative ANCA antibodies), connective tissue diseases (ANA, cardiolipine antibodies, anti -nRNP, Sm, SS-A, SS-B, Sci-70, Jo- 1), tuberculosis (direct bacterioscopy, automated radiometric BACTEC, the conventional culture on Lowenstein-Jensen (LJ) medium in pleural fluid, sputum and bronchoscopic fluid were negative), as well as all other potential background leading to the necrotic granuloma pathology type formation.

Based on immunofixation, monoclonal gammapathy was found. The amyloid depictions were confirmed undoubtedly in the kidney and heart samples with Congo staining and under polarized light on histopathological examination.

The manuscript was once again reviewed by a native speaker, member of language service board for  the Scientific Reports, Journal of Clinical Medicine and Vaccine journals.

Round 3

Reviewer 2 Report

The rebuttal of the authors is acceptable.

However, the confusing description of birefringent particles in the kidney (figure 3) must be fixed.

I have made many remarks on the manuscript (attached) and these must be addressed. 

Author Response

We are very grateful for any comments and the pdf file, which made the current revision of the manuscript much easier for us.

The reviewer rightly showed us where to make changes and such corrections have been made in the current manuscript.

Once again, we thank the reviewer for the time spent on developing and improving the text.